# A Review on Fast Direct Methods of Surface Integral Equations for Analysis of Electromagnetic Scattering from 3-D PEC Objects

**Ming Jiang [1,2,*], Yin Li [3], Lin Lei [1] and Jun Hu [1]**

[1] School of Electronic Science and Engineering, University of Electronic Science and Technology of China (UESTC), Chengdu 611731, China

[2] Institute of Electronic and Information Engineering, University of Electronic Science and Technology of China (UESTC), Chengdu 611731, China

[3] Department of Broadband Communication, Peng Cheng Laboratory, Shenzhen 518066, China

* Correspondence: jiangming@uestc.edu.cn

**Abstract:** This paper reviews a series of fast direct solution methods for electromagnetic scattering analysis, aiming to significantly alleviate the problems of slow or even non-convergence of iterative solvers and to provide a fast and robust numerical solution for integral equations. Then the advantages and applications of fast direct solution methods and the research trends are introduced in detail. Three different main methods are discussed, namely hierarchically off-diagonal low-rank matrices (HODLR) and skeletonization direct methods based on weak and strong admissibility condition. Numerical examples of computational complexity and electromagnetic scattering analysis of jet models are presented to demonstrate the efficiency and accuracy of each approach. Finally, a brief discussion is given on the main challenges and possible strategies of fast direct solution methods which still exist.

**Keywords:** integral equation; direct solver; skeletonization

## 1. Introduction

In the community of computational electromagnetic methods (CEM), surface integral equations (SIEs) have proven to be one of the powerful methods to analyze electromagnetic scattering and radiation problems, which is derived from surface equivalence principle. To numerically solve SIEs, the state-of-the-art Method of Moment (MoM) is typically used. Since the unknowns used to approximate the surface equivalence current are only distributed on the surface of the object, this can bring a lot of conveniences to discretization. To the solution of the dense matrix system of MoM, iterative methods based on Krylov subspace such as the generalized minimum residual (GMRES) and the conjugate gradient (CG) algorithms are applicable. The bottleneck arises due to the fact of the computational complexity and memory cost scale, as $O\left(N^2\right)$ or $O\left(N^3\right)$, where N is the number of unknowns. Therefore, it becomes prohibitive for large-scale modeling.

The emergence of many fast algorithms enable MoM to be used to solve real-world electromagnetic applications. Generally, these algorithms can be divided into two categories. The first are dependent on the kernel function of the problems of interest, namely Green's functions. The representative methods include the multilevel fast multipole algorithm (MLFMA) [1] and the fast Fourier transform (FFT) based method [2,3]. For instance, the MLFMA exploits the analytical harmonic expansion of Green's function and addition theorem, which can achieve O(NlogN) storage complexity and O(NlogN) computational complexity per matrix-vector multiplication for 3D PEC electromagnetic scattering problems. The FFT-based methods leverage the translation invariance property of the kernel function which can reduce memory requirement and CPU time complexity with $O\left(N^{1.5}\right)$ and $O\left(N^{1.5}logN\right)$ for 3D problems, respectively. The second are purely algebraic methods

that are kernel independent. These methods approximate some sub-blocks, with rank deficiency of the dense matrix, by low-rank algorithms such as the adaptive cross approximation (ACA) [4,5] and the multilevel matrix decomposition algorithm (MLMDA). Li et al. have provided an ample review on the low-rank algorithms for solving multi-scale problems in the electromagnetic field [6]. However, these fast algorithms based on iterative approaches always suffer from convergence issues, especially in the solution of multi-scale problems. Most of the iterative approaches are sensitive to the condition number of matrix systems and the number of iterations required to achieve a desired accuracy is highly problem-dependent. Although preconditioner techniques and domain decomposition methods (DDM) have been developed to cure a large majority of such problems, the convergence is still unpredictable. On the other hand, iterative solvers prove ineffective in solving multiple right-hand sides problems.

Recent exploration of fast direct solvers for integral equations can overcome the aforementioned deficiencies. The earliest research in the electromagnetic community can be traced back to the IES$^3$ algorithm proposed by Kapur and Long for the solution of the extraction of integral circuit structures [7]. At the same time, Michielssen and Boag also proposed a direct solution method for 2-dimensional slender smooth scatterers [8,9]. The compressed block decomposition (CBD) algorithm is presented by Herdring et al. for direct capacitance extraction with the help of the matrix decomposition algorithm and SVD technique [10]. A local-global solution (LOGOS) method was proposed to provide a useful strategy for modular electromagnetic analysis on large domains [11]. A fast multilevel direct solver based on a non-uniform sampling grid approach for electromagnetic scattering from a quasi-planar object is reported in [12]. Recently, it has been found that the multilevel nonuniform grid (MLNG) approach can be used for direct inversion of the electrical field integral equation (EFIE) [13]. In addition, a great deal of effort in applied mathematics has been devoted to the direct solution methods [14–22], which has also greatly inspired related research in the field of electromagnetics, such as the work of Greengard et al. It is well known that H-matrix is the most general class of hierarchical matrices, originally proposed by Grasedyck and Hackbusch [23,24]. In the structure of the H-matrix, the interaction between nearby groups is full-rank while the interaction between well-separated groups can be efficiently compressed as a low-rank format. This is based on the strong admissibility criterion. When the H-matrix is applied to discrete integral equations, it needs to satisfy the condition of the smoothing of the integral kernel. For the integral equations of an electromagnetic field, due to the oscillatory nature of its integral kernel, the H-matrix method can only be applied to the propagation, radiation and scattering problems in the mid-low frequency range. When the H-matrix is applied to high-frequency problems, it cannot achieve the acceleration effect of reducing storage and computational complexity [25]. In fact, any algebraically fast algorithms based on low rank compression are hard to speed up high-frequency problems. Because the rank of a matrix block is not a constant in a high frequency scenario, it is proportional to the dimension of the matrix block. Jiao completed some representative work [26–28] by applying the direct solution methods based on H-matrices and an advanced version of H$^2$-matrices. In [29], an H-matrix-based direct solver is used to accelerate the solution of the partial modification problem. The authors extend the direct solver based on skeletonization to solve SIEs for 3-D electrodynamic applications [30,31]. Guo et al. also analyzed and compared two direct methods of multi-level block inversion and multi-level LU decomposition [32]. Recently, Guo et al. have demonstrated a fast direct solver based on the butterfly method and randomized compression technique to achieve $O\left(N\log^2 N\right)$ CPU complexity and storage when applied to SIE solution of 3D PEC scattering problems. Furthermore, the butterfly method enhanced by the MPI-OpenMP parallel technique successfully solved irregular PEC target scattering problems with more than 10 million unknowns [33]. Fast direct solutions based on SIEs have also been developed for the simulation of penetrable objects. In [34], a quasi-block-Cholesky (QBC) algorithm exploring the checker-board symmetry pattern of the Poggio-Miller-Chang-Harrington-Wu-Tsai (PMCHWT) impedance matrix

was proposed to simulate human models. Recently, it has been reported that the multilevel matrix decomposition algorithm (MLMDA) based on the butterfly scheme [35] was also applied to homogeneous penetrable objects. The above investigation shows that the fast direct solution algorithm has always been a research hotspot in the field of computational electromagnetic fields. In this review, we will focus on our recent contributions in this area, which are based on three kinds of matrix structures, namely hierarchical off-diagonal low rank (HODLR), hierarchically semiseparable (HSS) matrix and $H^2$-matrix.

The remainder of the paper is organized as follows: in Section 2, the boundary value problem of PEC is formulated. A fast direct solver based on modified hierarchical off-diagonal low rank method (HODLR) is introduced in Section 3. Two kinds of direct solvers based on skeletonization factorization are illustrated in Section 4. Finally, a brief conclusion is given in Section 5.

## 2. Boundary Value Problem Statement

Consider electromagnetic scattering from an arbitrarily three-dimension (3-D) perfect electrically conducting (PEC) object $\Omega$. Time-harmonic electromagnetic fields $\left( \mathbf{E}^{inc}, \mathbf{H}^{inc} \right)$ impinge on the surface of object $\partial\Omega$ Then, the well-known electric field integral equation (EFIE) and magnetic field integral equation (MFIE) can be given by

$$- ik\eta \iint\limits_{\partial\Omega} \left( \mathbf{J}(r') + \frac{1}{k^2} \nabla \left( \nabla \cdot \mathbf{J}(r') \right) \right) G(r, r') dr' = \mathbf{E}^{inc}(r) \tag{1}$$

$$\frac{1}{2}\mathbf{J}(r') - \hat{\mathbf{n}} \times P.V. \iint\limits_{\partial\Omega} \mathbf{J}(r') \times \nabla G(r, r') = \hat{\mathbf{n}} \times \mathbf{H}^{inc}(r) \tag{2}$$

where $\mathbf{J}(r)$ denotes the induced surface current. $\hat{\mathbf{n}}$ is the outward unit normal to the surface of the object $\partial\Omega$. $G(r, r')$ is Green's function in 3-D free space, P.V. denotes Cauchy principal value integration, $k$ denotes the wavenumber and $\eta$ denotes the intrinsic impedance.

Usually, the combined field integral equation (CFIE) is employed to avoid internal resonances, which is the linear combination of EFIE and MFIE that is given by

$$\text{CFIE} = \alpha\text{EFIE} + (1 - \alpha)\eta\text{MFIE} \quad 0 \le \alpha \le 1 \tag{3}$$

where $\alpha$ is combination factor.

For the numerical solution of CFIE, the induced surface current $\mathbf{J}(r)$ is expanded in a series of N basis functions:

$$\mathbf{J}(r) = \sum_{n=1}^{N} I_n \mathbf{f}_n(r) \tag{4}$$

where $I_n$ denotes the expansion coefficients of surface electric current and $\mathbf{f}_n$ denotes Rao-Wilton-Glisson (RWG) basis function [36]. After the discretization and Galerkin testing, a N $\times$ N linear dense system of CFIE is obtained.

$$\mathbf{Z} \cdot \mathbf{I} = \mathbf{V} \tag{5}$$

We know that the solution of matrix Equation (5) via traditional direct methods such as Gaussian elimination or LU factorization is prohibitively expensive since the CPU and memory resources of these methods scale as $O\left(N^3\right)$ and $O\left(N^2\right)$, respectively. In the following sections, three kinds of fast direct solution of Equation (5) are introduced.

## 3. A Fast Direct Solver Based on Hierarchical Off-Diagonal Low Rank Method (HODLR)

Recently, a modified HODLR structure has been proposed to improve the capability of solving EM scattering problems [37]. The HODLR matrix structure is one special type of the H-matrix, in which all the off-diagonal matrices are compressed by low rank algorithms. Based on the idea, the original matrix can be factorized into the multiplication of several

block diagonal matrices. Finally, the Sherman–Morrison–Woodbury (SMW) formula [38] can be adopted to efficiently achieve the inverse of the block diagonal matrices.

Usually, the HODLR structure is constructed in the framework of a multilevel binary tree. The total basis functions are first decomposed into two groups, each containing half the total bases. If this kind of decomposition repeats $L-1$ times and the finest-level groups involve no more than $n_{min}$, a L-level binary tree will be constructed. For the group $i$ and group $j$ at a certain level $l$, the off-diagonal dense matrices $\mathbf{Z}_{ij}$ can be sparsely represented by low-rank compression methods, while the diagonal matrices should be further split into four smaller submatrices in the next level $L+1$. The process is performed $L-1$ times when all off-diagonal dense matrices are compressed by low-rank methods, while diagonal matrices at the finest level L are computed with MoM directly. Hence, the HODLR matrix with L levels can be factorized:

$$\mathbf{Z} = \begin{bmatrix} \begin{bmatrix} \mathbf{Z}_{11}^{(L)} & \mathbf{U}_{12}^{(L)}\mathbf{V}_{12}^{(L)} \\ \mathbf{U}_{21}^{(L)}\mathbf{V}_{21}^{(L)} & \mathbf{Z}_{22}^{(L)} \end{bmatrix} & \mathbf{U}_{12}^{(L-1)}\mathbf{V}_{12}^{(L-1)} & \cdots \\ \mathbf{U}_{21}^{(L-1)}\mathbf{V}_{21}^{(L-1)} & \begin{bmatrix} \mathbf{Z}_{33}^{(L)} & \mathbf{U}_{34}^{(L)}\mathbf{V}_{34}^{(L)} \\ \mathbf{U}_{43}^{(L)}\mathbf{V}_{43}^{(L)} & \mathbf{Z}_{44}^{(L)} \end{bmatrix} & \cdots \\ \vdots & \vdots & \ddots \end{bmatrix} \quad (6)$$

where, the superscripts denote the level and the subscripts denote the groups at the same level. The graphical representation of the three-level HODLR structure is shown in Figure 1.

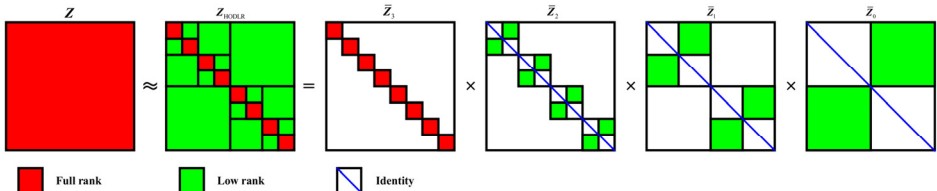

**Figure 1.** HODLR matrix structure with three levels and its factorization (L = 3) [37].

Next, $\mathbf{Z}$ can be cast into the multiplication of several diagonal block matrices after decomposition and compression, as shown in Figure 1. Mathematically, the HODLR structure can be factorized as

$$\mathbf{Z} = \overline{\mathbf{Z}}_L \overline{\mathbf{Z}}_{L-1} \cdots \overline{\mathbf{Z}}_1 \overline{\mathbf{Z}}_0 \quad (7)$$

$\overline{\mathbf{Z}}_L$ denotes the diagonal block matrix at the finest level, which is the original full rank matrix. The $i$-th diagonal blocks of $\overline{\mathbf{Z}}_1(l = 0, 1, \dots L-1)$ can be written as the following general form:

$$\overline{\mathbf{Z}}_{ii}^{(l)} = \begin{bmatrix} \mathbf{I} & \tilde{\mathbf{U}}_{2i-1,2i}^{(l+1)}\mathbf{V}_{2i-1,2i}^{(l+1)} \\ \tilde{\mathbf{U}}_{2i,2i-1}^{(l+1)}\mathbf{V}_{2i,2i-1}^{(l+1)} & \mathbf{I} \end{bmatrix} = \mathbf{I} + \tilde{\mathbf{U}}_{ii}^{(l)}\tilde{\mathbf{V}}_{ii}^{(l)} \quad (8)$$

Then, the inverse of Equation (8) can be easily obtained by the SMW formula:

$$\left( \overline{\mathbf{Z}}_{ii}^{(l)} \right)^{-1} = \mathbf{I} - \tilde{\mathbf{U}}_{ii}^{(l)} \left( \mathbf{S}_{ii}^{(l)} \right)^{-1} \tilde{\mathbf{V}}_{ii}^{(l)} \quad (9)$$

where

$$\mathbf{S}_{ii}^{(l)} = \mathbf{I} + \tilde{\mathbf{V}}_{ii}^{(l)}\tilde{\mathbf{U}}_{ii}^{(l)} \quad (10)$$

Therefore, the solution of the matrix equation can be obtained:

$$\mathbf{I} = \left(\overline{\mathbf{Z}}_0\right)^{-1} \left(\overline{\mathbf{Z}}_1\right)^{-1} \cdots \left(\overline{\mathbf{Z}}_{L-1}\right)^{-1} \left(\overline{\mathbf{Z}}_L\right)^{-1} \mathbf{V} \tag{11}$$

However, in the traditional HODLR structure, all off-diagonal matrices are simply compressed by low-rank approximation techniques. If two sets of bases are not far enough, the corresponding interaction matrix does not have the low-rank property and the rank remains large after the compression. In addition, the dimensions of off-diagonal matrices might be large, so the direct and simple processing in this traditional method is very inefficient or even impractical.

In [37], instead of applying the low rank compression method to the off-diagonal block matrices directly, the modified technique subdivides the original large off-diagonal blocks into smaller ones. This subdivision process is determined by the extended admissibility condition (EAC). Compared with the admissibility condition, such an approach can further reduce the ranks of off-diagonal blocks. The computational resource can be reduced in terms of both computational cost and memory storage cost.

Firstly, we consider a PEC sphere (r = 1.0 m) to demonstrate the complexity of both computational time and storage cost [37]. With the increasing of the frequency of the plane wave from 0.6 GHz to 3.0 GHz, the number of unknowns is varied from 18,252 to 458,901. The tolerance of ACA and SVD is fixed at 0.001 in this example. Figure 2a shows that the modified solver and the conventional solver have a complexity of $O\left(N^{1.5}\right)$ and $O\left(N^2\right)$ in terms of matrix assembly, while the complexity of factorization and storage are $O\left(N^2\right)$ and $O\left(N^{1.5}\right)$ for both solvers as shown in Figure 2b,c. However, we can see that the cost of factorization and the memory is less due to the smaller rank of the modified solver. The efficiency can be improved significantly by the modified solver.

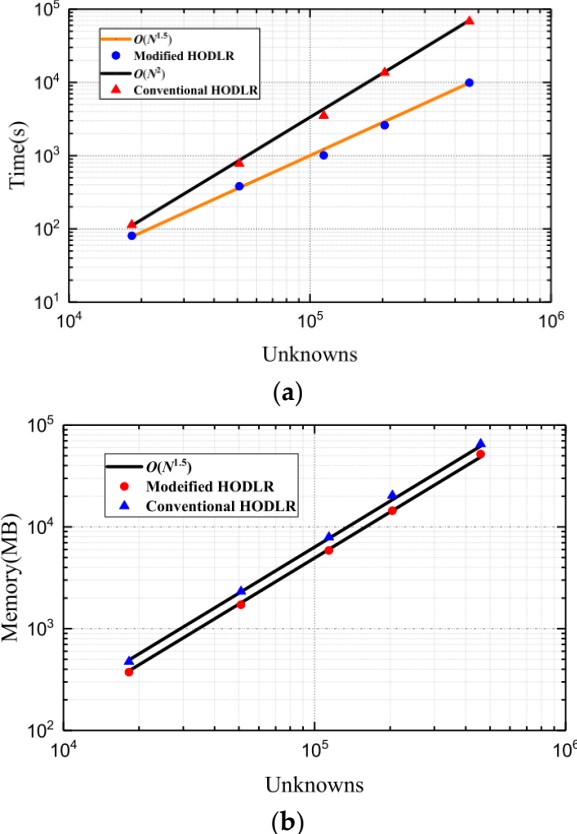

**Figure 2.** *Cont.*

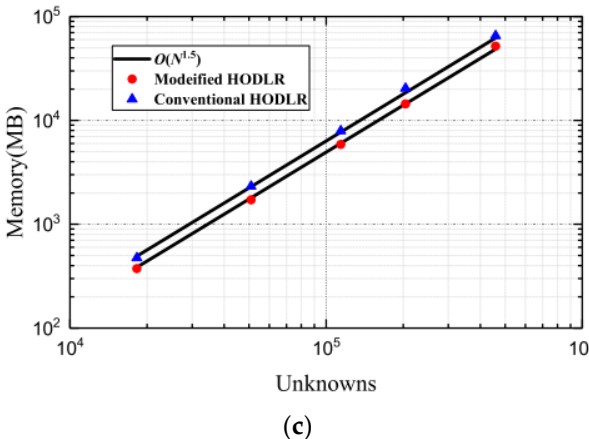

(c)

**Figure 2.** Complexity analysis of the modified and conventional HODLR solver [37]. (**a**) Computational time for compression. (**b**) Computational time for factorization. (**c**) Memory cost of the inverse matrix.

To show the accuracy of the modified HODLR method, a PEC sphere with radius $r = 3.0\lambda$ is considered, which is discretized into 37,647 RWGs and an 8-level matrix structure is constructed. The results of the proposed method are compared with those of the Mie series. It is shown that a good agreement is achieved in Figure 3.

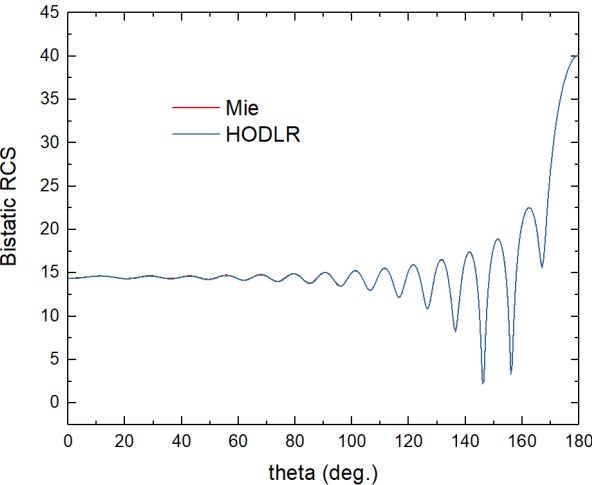

**Figure 3.** Bistatic RCS (HH-polarized) of the sphere with radius $r = 3.0\lambda$.

## 4. Skeletonization-Based Direct Solvers

Recently, the skeletonization-based fast methods [15–17] have attracted much attention in the area of CEM. The matrix structures of these solvers are based on weak admissibility condition, which is similar to the hierarchically semiseparable (HSS) [16] matrix. The basic idea behind skeletonization is that the interactions between well-separated groups are calculated by dominant bases named skeletons, which are selected via the interpolative decomposition (ID) [39]. After obtaining the hierarchical low-rank representation, the inverse of the system matrix can be performed recursively [17].

Skeletonization factorization begins with building a multilevel structure with octree technique. Here, we assume that the off-diagonal matrix $\mathbf{Z}_{ij}$ represents interaction between groups $i$ and $j$, which is of low rank property and compressed by ID:

$$\mathbf{Z}_{ij} = \mathbf{L}_i \mathbf{S}_{ij} \mathbf{R}_j \tag{12}$$

where, $\mathbf{L}_i$ and $\mathbf{R}_j$ denote the projection matrices associating with $i$ and $j$. $\mathbf{S}_{ij}$ is the subset of $\mathbf{Z}_{ij}$, which physically means the skeleton interaction matrix between groups $i$ and $j$. The explicit expression of $\mathbf{L}_i$ can be written as:

$$\mathbf{L}_i = \mathbf{P}_i\left(\mathbf{I}, \mathbf{T}_i^T\right)^T \tag{13}$$

$\mathbf{P}_i$ is the permutation matrix. $\mathbf{T}_i$ is the interpolation matrix related to the skeleton and redundant basis functions.

How to efficiently extract the skeleton bases of one group and its projection matrix is the key to the method. One should regroup all interaction matrices associated with the chosen group into a new matrix. However, the size of such a matrix is too large, so it would be costly to construct such a dense matrix. To cut down the high cost of constructing the matrix, a proxy surface approach based on Huygens' principle is generally used. The interactions among the chosen group and the groups outside the proxy surface can be replaced with the interactions between the chosen group and the proxy surface. In this way, skeletons of the chosen group can be efficiently extracted by the combination of neighboring interaction matrices and proxy interaction matrices, i.e.,

$$\mathbf{Z}_i = \left(\mathbf{Z}_{i,nb}, \mathbf{Z}_{i,s}, \mathbf{Z}_{nb,i}^T, \mathbf{Z}_{s,i}^T\right) \tag{14}$$

The size of $\mathbf{Z}_i$ can be shrunk from $2(\mathrm{N} - n_i) \times n_i$ to $n_i \times 2\left(n_i^{nb} + 2n_i^s\right)$, $n_i^s$ is the number of sample points on the proxy surface and $n_i^{nb}$ is the number of basis functions in the neighboring groups of the chosen group $i$.

After off-diagonal blocks have been compressed, the original impedance matrix $\mathbf{Z}$ can be represented as:

$$\mathbf{Z} = \mathbf{D} + \mathbf{LSR} \tag{15}$$

Then, the inverse of $\mathbf{Z}$ can be written as:

$$\mathbf{Z}^{-1} = \tilde{\mathbf{D}} + \tilde{\mathbf{L}}(\mathbf{E} + \mathbf{S})^{-1}\tilde{\mathbf{R}} \tag{16}$$

In [40], we developed a fast direct solver based on recursive skeletonization factorization (RSF) [41], based in turn on weak admissibility condition. To accelerate the procedure of skeletonization, a novel skeletonization strategy is proposed. Firstly, a series of equivalence points are generated on the sphere proxy surface. Two constant vector basis functions are defined on each point with $\hat{\theta}$ and $\hat{\varphi}$ components, which represent electric and magnetic currents on the proxy surface. In terms of Huygens' principle, the interactions between the group and its far field can be expressed by the interactions between the group and these sampled points. Compared with the conventional method, these points can be uniformly distributed on the proxy surface. For example, in our empirical and numerically test, the sample number P can be defined as $\mathrm{P} = \max\left\{\left(\frac{kr_i}{3} + 3\right), 5\right\}$, which increases proportionally with the radius of proxy sphere. Besides, the skeletons in the selected neighboring groups are further used to reduce the number of basis functions of the neighboring groups, since they are dominant bases that can represent the interactions of both near-field and far-field. Finally, the size of proxy matrix is further shrunk and the cost of implementation would be reduced.

Once the skeletonization factorization has been performed level by level and all relevant matrices in each group have been obtained, the RSF can be applied to the compressed form of $\mathbf{Z}$ in Equation (5).

$$\mathbf{Z} = \tilde{\mathbf{V}}_L \ldots \tilde{\mathbf{V}}_1 \mathbf{Z}_0 \tilde{\mathbf{W}}_1 \ldots \tilde{\mathbf{W}}_L \tag{17}$$

Finally, the solution of Equation (17) can be obtained by a series of matrix-vector multiplications:

$$\mathbf{I} = \mathbf{Z}^{-1}\mathbf{V} = \left(\tilde{\mathbf{W}}_L\right)^{-1}\cdots\left(\tilde{\mathbf{W}}_1\right)^{-1}(\mathbf{Z}_0)^{-1}\left(\tilde{\mathbf{V}}_1\right)^{-1}\cdots\left(\tilde{\mathbf{V}}_L\right)^{-1}\mathbf{V} \tag{18}$$

The inverse of $\tilde{\mathbf{V}}_1$ and $\tilde{\mathbf{W}}_1$ can be directly obtained during the skeletonization and only the diagonal matrix $\mathbf{Z}_0$ needs to be inverted by LU decomposition, which implementation is detailed in [40].

In order to demonstrate the computational complexity and storage requirement of the proposed skeletonization method, a radius r = 2.0 m PEC sphere is analyzed in [40]. The frequency of incident plane wave is set to 300 MHz, 600 MHz, 1200 MHz and 2400 MHz. Mesh size is fixed at $0.1\lambda$. In Figure 4, it is validated that the skeletonization factorization time and memory costs scale with $\mathrm{O}\left(\mathrm{N}^{1.8}\right)$ and $\mathrm{O}\left(\mathrm{N}^{1.3}\right)$, respectively.

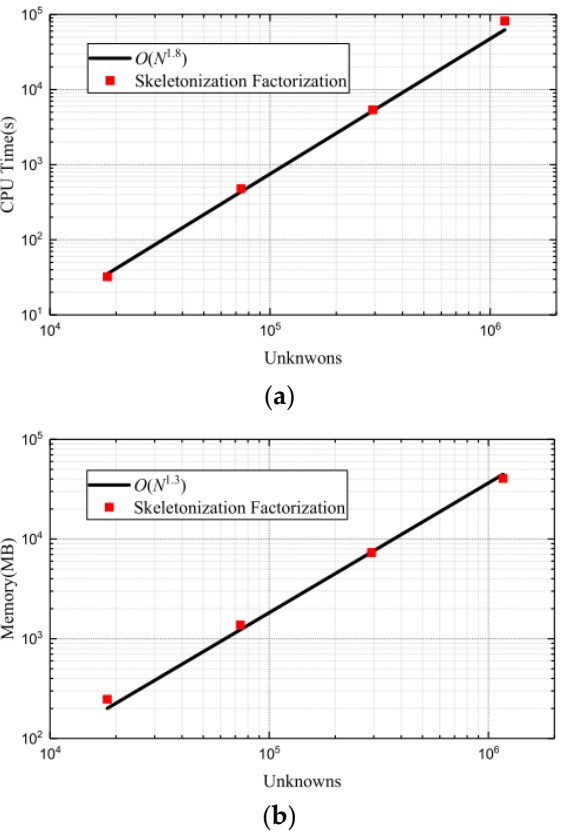

**(a)**

**(b)**

**Figure 4.** Complexity of computational time and memory costs of the skeletonization solver [40]. (**a**) Complexity of factorization. (**b**) Memory costs of the inverse matrix.

The aforementioned skeletonization factorization is based on weak admissibility (WASF). The main process is that all interactions involving far-field interactions and neighboring interactions contributed from redundant bases are decoupled, so zero matrices are introduced in the relevant blocks, while all dominating interactions between skeletons remain and will be further aggregated into a coarser level for further factorization. Consequently, a series of block diagonal matrices is used to achieve the multiplicative decomposition representation of the original system matrix. The inverse can be solved easily. However, since neighboring interactions need to be assembled in advance in the skeletonization factorization and the resultant ranks remain large, it requires much memory and time consumption for multilevel compression.

Therefore, we further proposed the skeletonization factorization based on strong admissibility condition (SASF) [42,43]. Only the far-field interactions are compressed instead of both the far-field and neighboring interactions. Similarly, the proxy surface method based on Huygens' surface is used. The data structure built in SASF is similar to MLFMA, which is a kind of $H^2$-matrix. Subsequently, the inverse of system matrix can be also obtained by the recursive skeletonization factorization.

In [42], we demonstrated that the complexity of computational time and memory costs of SASF solver are with $O\left(N^{1.5}\right)$ and $O(N\log N)$ for electrically moderate problems, respectively. In order to compare with the accuracy of the two kinds of direct solvers based on skeletonization, a model with length 1.75 m, width 0.9 m and height 0.2 m is investigated, as shown in Figure 5. The plane wave at 3.0 GHz is illuminated from the nose of the model with z-direction polarization. The mesh size is 0.1λ and 65,814 RWGs are needed. Here, ID tolerance of both WASF and SASF is set as 0.001. In Figure 5, the Monostatic RCS results in x-y plane are compared among SASF, WASF and MLFMA. They agree with each other very well.

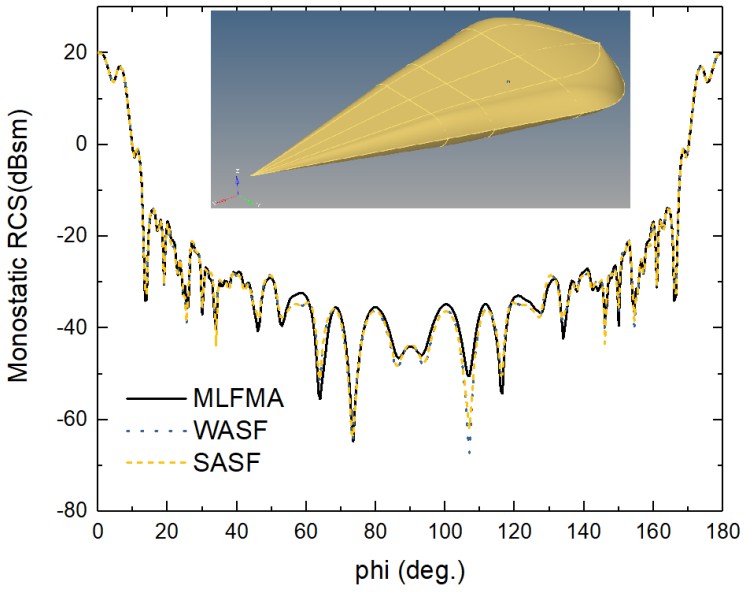

**Figure 5.** Monostatic RCS (HH-polarized) of the model at 3.0 GHz.

## 5. Conclusions

In this paper, three kinds of fast direct solution algorithms are investigated to solve surface integral equations for 3-D electromagnetic wave scattering from PEC targets. The modified HODLR algorithm can further compress the rank-deficient off-diagonal blocks effectively compared with the conventional HODLR method. Then, two kinds of skeletonization-based direct solvers are introduced, which are based on weak and strong admissibility condition and utilize a recursive skeletonization factorization process. However, skeletonization factorization leads to a relatively high rank since all off-diagonal blocks are performed by compression techniques, which implies that all near-field of nearby-groups are assumed to be low-rank in nature. Therefore, the bottom-up factorization process is inefficient, while SASF extracts fewer skeleton basis functions because only far-field groups that satisfy the strong admissibility condition are considered to be compressed. Meanwhile, it is efficient to compress the far-field interaction matrix with the reduced dimension in the SASF. Recently, the SASF approach has been extended to apply to the solution of homogeneous penetrable objects [43], but there are few related reports on the direct solution algorithm for solving the PEC-dielectric composite problem. Finally, the proposed direct solvers provide a promising, alternative tool for DDM when facing complex electromagnetic problems with multiscale geometrical features [44].

**Author Contributions:** M.J. and J.H. conceived of the presented idea. Y.L. verified the analytical methods. L.L. performed the numerical calculations and performed the numerical simulations. All authors discussed the results and contributed to the final manuscript. All authors have read and agreed to the published version of the manuscript.

**Funding:** This work was supported in part by National Natural Science Foundation of China under Grant No. 62071103, 61871090. Guangdong Basic and Applied Basic Research Foundation under grant 2019A1515111166, 2022A1515010483. This work was also supported in part by The Major Key Project of PCL Department of Broadband Communication.

**Conflicts of Interest:** The authors declare no conflict of interest.

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
