# Peer review of "A Review on Fast Direct Methods of Surface Integral Equations for Analysis of Electromagnetic Scattering from 3-D PEC Objects"

_electronics, doi:10.3390/electronics11223753_

Round 1

Reviewer 1 Report

As stated by the authors: "This paper reviews a series of fast direct solution methods for electromagnetic scattering analysis...". However, this paper is actually a compilation of the papers published by the authors (see the attached documents). Even being a review of previous research, the paper should not be, more or less, a simple reproduction of existing publications, containing the same figures, graphs, equations and text. Besides this, my opinion is that the paper is not in the scope of the journal, as well as the special issue, and that would be an additional reason for not accepting of this paper for publication in the present form. Moreover, its contribution to the field and the importance of the results presented is not significant. Most likely, the paper would not be attractive to the potential readers in the field.

There are also some other shortcomings:

1. references were not numbered in the order of appearance in the text,

2. text in lines 171 to 173 should be deleted,

3. formatting of the text is incorrect in some parts, such as in lines 32, 33 and other in page 1,

4. background of the problem, aim, new results and adequately supported conclusions are missing.

Reviewer 2 Report

The manuscript discusses various fast methods to accelerate the direct (matrix inversion) methods for solution of the surface integral equation in electromagnetic scattering by PEC targets. Overall, the topic is relevant and a concise review of it is given. The manuscript is suitable for Electronics, but the following issues must be addressed before the publication.

1)  I recommend to replace “integral” by “surface-integral” in the title and to mention PEC, since that is the main topic of the review.

2)  The Introduction nicely sets the stage for the review, by discussing the acceleration of both iterative and direct methods. But it relates mostly to surface-integral methods. The only mention of volume-integral methods (line 98) appears unexpectedly and feels unfinished. The authors should either extend the discussion of volume-integral methods a little bit, or state explicitly that only a single paper, accelerating volume-integral methods, exist.

3)  The authors should also explicitly discuss penetrable homogeneous objects (e.g., dielectrics). While these seems to be outside of the main subject of the manuscript, it should be discussed at least in Introduction. A general question is whether the methods, discussed throughout the manuscripts (for PEC) can be adapted to the case of penetrable objects.

4)  When performing the test simulations, the authors discuss simulation time, but almost ignore the accuracy. Most fast methods increase speed at the cost of accuracy, so it is important to discuss the latter as well. Right now, there is only a single figure (Fig.4), which shows that accuracy is good enough, but no numbers are given. Moreover, it is, for instance, possible to improve the speed of the standard method by decreasing the number of unknowns (increasing the size of surface elements) – is it inferior to the acceleration proposed by the authors? Overall, speed itself is not relevant, instead one should compare the computational time required to reach a certain prescribed accuracy.

5)  There are a lot of English errors. Some are specified below, but the authors are highly encouraged to thoroughly check the whole manuscript once again.

·       Lines 171-173 should be removed.

·       Line 242: “. One” -> “, one”

·       Line 284: “inversed” -> “ inverted”

·       Lines 284-285: “which are detailed implemented in” -> “, which implementation is detailed in”

·       Line 292: “Unknwons” -> “Unknowns”

·       Line 310: “thus” -> “Thus”.

Reviewer 3 Report

The paper appears to be a useful review of direct methods to solve dense matrix algebraic equations coming from the Combined Field Integral Equation (CFIE)  applied to electromagnetic scattering from perfectly conducting objects.

Some sentences require a revision from a native English speaker. Some minor criticisms follow.

a)       The bold dot operator is used in several formulas with different meanings. In (1) the dot denotes the scalar product between vectors. In (5), (10), (11) and other formulas the dot is used to denote matrix-vector products; in my opinion these dots could be omitted.

b)      In row 243 the symbol I should be changed into i.

Reviewer 4 Report

The question is how the authors took into account the shape of the object in the calculation of the RSC. Fig. 4 shows the dimensions of the rectangle modeling the tested object

Round 2

Reviewer 1 Report

The authors have made minor changes in the paper. They did not change the parts of the text (including figures) that have been repeated from the already published papers. Still, I do not recommend this paper for publication.

Author Response

We sincerely thank the editors and the reviewers for their constructive comments and helpful discussions that have contributed much to improve the clarity and presentation of the manuscript. Based on these comments, we have made corresponding revisions. All the comments are responded in the following, and the revisions are highlighted in the revised manuscript. We hope the revision is in a better status for the reviewers' and editors' further consideration.

#Reviewer: 1

Comments for Transmittal to Author

The authors have made minor changes in the paper. They did not change the parts of the text (including figures) that have been repeated from the already published papers. Still, I do not recommend this paper for publication.

Response:

Thank you for your suggestion

In our revised manuscript, we have updated two numerical examples (Figure 3 and Figure 5).

Reviewer 2 Report

The authors have significantly improved the manuscript. I have only two minor comments:

·       Line 3: “analysis electromagnetic” -> “analysis of electromagnetic”

·       Line 280: “numerical calculations” and “numerical simulations” seem to be duplicates.

Author Response

Q1.  The authors have significantly improved the manuscript. I have only two minor comments:

  • Line 3: “analysis electromagnetic” -> “analysis of electromagnetic”
  • Line 280: “numerical calculations” and “numerical simulations” seem to be duplicates.

Response:

Thank you very much for pointing out these issues. We hope the revised manuscript becomes in a better status for your further consideration.  

Round 3

Reviewer 1 Report

I have no additional comments.